# Glycoprotein Non-Metastatic Protein B: An Emerging Biomarker for Lysosomal Dysfunction in Macrophages

**DOI:** 10.3390/ijms20010066

**Published:** 2018-12-24

**Authors:** Martijn J.C. van der Lienden, Paulo Gaspar, Rolf Boot, Johannes M.F.G. Aerts, Marco van Eijk

**Affiliations:** Leiden Institute of Chemistry, Leiden University, 2333 CC Leiden, The Netherlands; m.j.c.van.der.lienden@lic.leidenuniv.nl (M.J.C.v.d.L.); paulo.gaspar@insa.min-saude.pt (P.G.); r.g.boot@LIC.leidenuniv.nl (R.B.); j.m.f.g.aerts@LIC.leidenuniv.nl (J.M.F.G.A.);

**Keywords:** GPNMB, lysosomal storage disorders, metabolic syndrome, phagocytosis, autophagy, lysosome, macrophage, foam cell, inflammation, MITF

## Abstract

Several diseases are caused by inherited defects in lysosomes, the so-called lysosomal storage disorders (LSDs). In some of these LSDs, tissue macrophages transform into prominent storage cells, as is the case in Gaucher disease. Here, macrophages become the characteristic Gaucher cells filled with lysosomes laden with glucosylceramide, because of their impaired enzymatic degradation. Biomarkers of Gaucher cells were actively searched, particularly after the development of costly therapies based on enzyme supplementation and substrate reduction. Proteins selectively expressed by storage macrophages and secreted into the circulation were identified, among which glycoprotein non-metastatic protein B (GPNMB). This review focusses on the emerging potential of GPNMB as a biomarker of stressed macrophages in LSDs as well as in acquired pathologies accompanied by an excessive lysosomal substrate load in macrophages.

## 1. Inherited Lysosomal Storage Disorders

Lysosomal storage disorders (LSDs) comprise at least fifty distinct disorders, each caused by specific defects in the function of the lysosomal apparatus [1,2]. In LSDs, primary and secondary metabolites accumulate within lysosomes of specific cells, which in turn gives rise to progressive multi-organ pathologies. In many LSDs, tissue macrophages are among the prominent storage cells. Of note, with each particular LSD the clinical manifestation is heterogeneous, resulting in neonatal, infantile, juvenile, and adult variants. This heterogeneity is thought to stem from different primary genetic defects impacting differently on residual activity of a lysosomal enzyme. However, complex interplay between the genetic defect, modifier genes, epigenetics, and environmental factors seems to further contribute to variable clinical manifestation. This is exemplified by Gaucher disease (GD), a relatively common LSD [3]. GD is caused by an inherited deficiency in the lysosomal β-glucosidase glucocerebrosidase (GBA), causing an accumulation of its substrate glucosylceramide (GlcCer) [4]. GlcCer is the simplest glycosphingolipid, consisting of a glucose linked to the lipid moiety ceramide [5]. Lysosomal GlcCer storage occurs in GD patients almost exclusively in tissue macrophages, thus transforming into Gaucher cells [6]. The accumulation of viable Gaucher cells in tissues is thought to contribute to characteristic symptoms of adult GD patients, such as the enlargement of liver and spleen, anemia, and skeletal deterioration [3,7]. The overall severity of GD may vary considerably among patients, and consequently, different phenotypic variants are historically distinguished: the collodion baby with impaired skin permeability features incompatible with life outside the womb, the acute (infantile, type 3) and sub-acute (juvenile, type 2) variants with fatal neurological symptoms and the non-neuronopathic (adult, type 1) variant most common in Caucasian populations [3,7]. There is no strict correlation between mutations in GBA and disease manifestation in GD patients [8,9]. The most striking illustration of this comes from reports on monozygotic GD twins with marked discordance in symptoms [10,11]. The remarkable poor predictive value of the GBA genotype for GD phenotype complicates confirmation of diagnosis. Currently, the clinical assessment of Gaucher patients includes analysis of blood parameters (platelet count), examination of inflicted liver and spleen (MRI)/computed tomography (CT), skeletal status (MRI/X-ray), and a quality-of-life survey [3,12,13]. As described below, the demonstration in plasma of biomarkers (i.e., metabolites or proteins specifically secreted by the lipid-laden macrophages (Gaucher cells)), provides an additional tool to confirm the diagnosis of GD and may assist the monitoring of progression of disease [14]. Such biomarkers are also increasingly exploited to assess responses to costly therapies based on chronic intravenous supplementation with macrophage-targeted recombinant GBA or a pharmacological reduction of endogenous GlcCer by oral administration of inhibitors of glucosylceramide synthase [7,15].

## 2. Gaucher Cell Biomarkers: Lipids

Since the Gaucher cells primarily accumulate GlcCer, plasma glycosphingolipid abnormalities in GD patients have received considerable interest. Plasma of symptomatic GD-patients was found to show only moderately elevated levels of GlcCer, being associated with lipoproteins [16]. Likely, the excessive GlcCer in the patient’s plasma does not stem from Gaucher cells, but rather from hepatocytes. The same may hold for the elevated ganglioside GM3 observed in plasma of GD patients [17]. There is consensus that plasma GlcCer has no value as a GD biomarker. More relevant in this connection is the occurrence of the more than hundred-fold increased glucosylspingosine (GlcSph) in plasma of GD patients and animal models of GBA deficiency [18,19]. GlcSph is de-acylated GlcCer lacking the fatty acyl moiety. This sphingoid base was demonstrated to be actively formed inside lysosomes by the enzyme acid ceramidase acting on accumulating GlcCer [20]. Intralysosomally formed GlcSph may partly leave cells, and even leave the body via bile and urine. The prominent cellular producers of plasma GlcSph in GD patients seem to be visceral Gaucher cells [18], however many cell types produce GlcSph during marked GBA deficiency. Indeed, about ten-fold increased plasma GlcSph was observed in plasma of patients with Action Myoclonus Renal Failure syndrome (AMRF) [21]. This disorder is caused by genetic deficiency of lysosome membrane protein 2 (LIMP-2; also called Scavenger Receptor Class B Member 2 (SCARB2)), the membrane protein involved in transport of newly formed GBA to lysosomes [22]. GBA is markedly reduced in many cell types of AMRF patients, but actually not in their macrophages, likely due to some alternative transport mechanism in these cells or their ability of re-uptake of faulty secreted GBA by other cells [23]. At present, plasma GlcSph is considered to be useful GD biomarker and its measurement is already broadly used [18,19,24,25]. Of note, sphingoid bases, rather than the corresponding primary storage lipids, are also used as markers in other sphingolipid storage disorders [7,26]. Examples are galactosylsphingosine in Krabbe Disease, globotriaosylsphingosine in Fabry Disease, a phosphorylcholinesphingosine (lyso-sphingomyelin 509) in Niemann-Pick type C (NPC) and B (NPB) [27,28,29]. Convenient and sensitive multiplex measurements of several sphingoid bases have been developed and their use may assist in the confirmation of diagnosis of several sphingolipid storage disorders [30,31,32]. A role of the sphingoid bases in pathophysiology has also been hypothesized. For example, it has been proposed that excessive GlcSph may play a role in abnormal osteoblast differentiation and thus contribute to osteoporosis in GD patients [33]. A role of GlcSph as auto-antigen has been identified, promoting B-cell proliferation and the associated risk for multiple myeloma, a common cancer in GD patients [34,35]. Recently, it was reported that chronic administration of GlcSph to mice induces organomegalies and hematological abnormalities characteristic of GD [36]. Furthermore, excessive GlcSph has been proposed to promote alpha-synuclein aggregation [37]. This may provide an explanation for the increased risk of individuals with abnormal GBA to develop Parkinson’s disease [38]. Likewise, excessive globotriaosylsphingosine (lyso-Gb3) in Fabry patients is thought to contribute to neuronopathic pain and loss of podocytes [39,40]. It is of interest to point out that apparently a dysfunction in lysosomal catabolism of glycosphingolipids leads to metabolic adaptations, generating secondary metabolites that ultimately may cause specific symptoms beyond the storage cells [41]. A recently recognized glycolipid abnormality in GD patients concerns glucosylcholesterol (GlcChol): it appears that glucosylcholesterol is formed in cells by sequential action of the enzymes glucosylceramide synthase (GCS) and the transglucosylating non-lysosomal GBA variant GBA2 [42]. Lysosomal glucocerebrosidase (GBA) normally degrades GlcChol, but during lysosomal cholesterol accumulation the enzyme forms via transglucosylation of cholesterol GlcChol, using GlcCer as a glucose donor [42,43]. This pathway explains the massive increase in GlcChol in livers of mice with NPC, a condition caused by defects in either Npc1 or Npc2, proteins involved in the normal efflux of cholesterol from lysosomes [42]. Currently, biochemical confirmation of the diagnosis of NPC relies on the identification of cholesterol accumulation in patient-derived fibroblasts and the measurement of excessive plasma oxysterols by advanced mass spectrometry [44,45]. Oxysterols are formed in the body through enzymatic and non-enzymatic reactions involving reactive oxygen species (ROS). The latter reaction seems to be driving the enhanced levels of oxysterols in NPC [45,46,47,48,49]. Moderate elevation of oxysterol levels is also observed in other cholesterol-related storage diseases, such as atherosclerosis, obesity, and diabetes [50,51,52]. The role of GlcChol in pathophysiology of NPC still warrants investigation. Of note in this connection, the pharmacological inhibition or genetic deletion of GBA2 causing a marked reduction of GlcChol has been found to ameliorate disease manifestations in NPC mice [53]. Furthermore, N-butyl-1-deoxynojirimycin (Zavesca or Miglustat), an inhibitor of GCS and GBA2, is an approved drug to treat the neurological symptoms of NPC [54,55,56,57].

## 3. Gaucher Cell Biomarkers: Proteins

The discovery of protein markers of Gaucher cells was prompted by the development of enzyme replacement therapy (ERT) for non-neuropathic GD some three decades ago by researchers at the National Institutes of Health [58]. Brady and colleagues used GBA isolated from human placentas which was being modified in its N-glycans to favor mannose receptor-mediated uptake by macrophages following intravenous administration [59]. This macrophage-targeted ERT was found to result in prominent corrections in organomegaly and hematological symptoms of GD patients [60]. The high costs associated with ERT of GD patients limited its application and stimulated research on personalized ERT, that is, the minimal effective dose of the recombinant enzyme for each patient [61,62]. Novel tools to sensitively monitor corrections in Gaucher cell burden of GD patients following ERT became urgently needed. Already reported were a number of plasma protein abnormalities in Gaucher patients, for example elevated levels of lysozyme, beta-hexosaminidase, ferritin, tartrate-resistant acid phosphatase (TRAP), and angiotensin-converting enzymes (ACE), (see [63] for a review. However, for none of these abnormalities it was clear that they are uniquely related to Gaucher cells and not also released by other cell types, as for example TRAP by pro-inflammatory macrophages, osteoclasts, and dendritic cells [64]. Subsequent research led to the discovery that Gaucher cells massively produce and secrete the enzyme chitotriosidase (CHIT1), causing a stunning average 1000-fold elevated plasma level in type 1 GD patients [65]. CHIT1 has been subsequently studied in great detail [65,66,67,68,69,70,71,72,73,74]. Importantly, it was found that the enzyme is specifically produced in tissue macrophages and neutrophils. In particular, Gaucher cells are producers of CHIT1 that is partly routed to lysosomes and partly secreted [68,71]. Improved substrates were next developed to accurately monitor CHIT1 levels in plasma of patients [75,76]. Plasma CHIT1 has been extensively investigated in relation to GD in clinical centers applying ERT. From these studies, it has become apparent that the reductions in plasma CHIT1 of GD patients following ERT have a prognostic value for corrections in organomegaly and the risk for long-term complications [77]. Of note, elevated plasma CHIT1 is not unique for GD [73]. The enzyme levels may be increased during various disease conditions, albeit to a much lesser extent as in type 1 GD patients [78,79,80]. Many LSDs show modest elevations in plasma CHIT1, most notably Fabry Disease and NPC [81,82,83]. Likely, the accumulation of materials in lysosomes of macrophages induces expression of CHIT1. A major drawback regarding CHIT1 as marker stems from the common occurrence of a duplication in the *CHIT1* gene causing absence active CHIT1 [67]. Homozygosity for this mutation occurs relatively frequently, being present in about 1 in 20 individuals in most ethnic groups. CHIT1 deficiency also occurs with the same frequency among GD patients [67]. This stimulated a search for additional protein markers of Gaucher cells. It was subsequently discovered that chemokine (C-C motif) ligand 18 (CCL18), also called pulmonary and activation- regulated chemokine (PARC) is also massively produced and secreted by Gaucher cells, resulting in twenty to forty-fold elevated plasma levels [84,85,86]. Corrections in plasma CCL18 and CHIT1 during ERT mimic each other closely, illustrating the common source of these markers being the Gaucher cell [85]. Like CHIT1, CCL18 is also elevated in NPC patients [87,88,89]. Monitoring corrections in plasma CHIT1 and/or CCL18 is not only performed in patients receiving ERT for which presently multiple recombinant enzymes are registered [90,91]. Corrections of Gaucher cell markers are also monitored in GD patients treated by means of substrate reduction therapy (SRT). In this alternative therapeutic approach, an inhibitor of GCS is orally administered to GD patients to reduce the endogenous synthesis of GlcCer and thus balance the impaired capacity of lysosomal degradation of the lipid [41]. Registered for SRT of type 1 GD are at present two GCS inhibitors miglustat and eliglustat [92,93,94]; responses in CHIT1, CCL18 and GlcSph to the SRT therapies have been analyzed [95]. 

## 4. Emerging Marker: Glycoprotein Non-metastatic Protein B (GPNMB) 

In recent years, the impact of GBA deficiency is increasingly studied in mouse models, either generated by genetic modification or pharmacologically induced with GBA inhibitors. The two existing protein biomarkers of storage macrophages in GD patients are unfortunately of no use for these murine GD models. In the mouse, CHIT1 is not expressed by phagocytes due to a different promotor [73]. In addition, no rodent homologue of CCL18 exists [85]. Moran et al. studied differentially expressed transcripts in a type 1 GD spleen [84]. Among the observed overexpressed mRNAs was the one coding for glycoprotein non-metastatic protein B (GPNMB). GPNMB was previously shown to be induced upon stimulation of monocytes with granulocyte-macrophage colony stimulating factor (GM-CSF) as well as with M-CSF [96]. Much later, Kramer and colleagues observed in their analysis of the proteome of normal and GD spleens marked increases in GPNMB in patients’ tissues [97]. The isolation of Gaucher cells by laser-capture revealed the massive presence of the protein in Gaucher cells. Moreover, the release of a soluble fragment of GPNMB was observed, explaining the up to several hundred-fold elevated levels in plasma of GD patients, as can be detected by ELISA [97]. Furthermore, it became apparent that also GBA-deficient mice in the hematopoietic lineage that form Gaucher cells show elevated GPNMB [98]. Treatment of such mice by substrate reduction therapy as well as lentiviral gene therapy leads to prominent corrections in GPNMB in key organs [97,98,99]. Independently, other researchers noted in other non-neuronopathic GD mouse models increased expression of GPNMB [33,100]. Zigdon and co-workers reported elevated GPNMB in cerebrospinal fluid (CSF) of type 3 GD patients and a pharmacological neuronopathic GD mouse model [101]. In a larger GD cohort, the applicability of GPNMB as biomarker was carefully examined [102]. This study revealed a correlation between serum GPNMB levels and disease severity [102].

Macrophage storage, reflected by a foamy cell appearance, is also observed in NPC. Interestingly, in NPC mouse models it was demonstrated that these macrophages (Iba1^+^ cells) showed high GPNMB protein levels in spleen, liver, and brain [103]. These observations extend on the earlier reported gene expression elevations in the same tissues in NPC mouse models [104,105]. Furthermore, GPNMB was found to be elevated in human NPC plasma samples, correlating with CHIT1 levels [103]. In summary, like CHIT1, GPNMB is strongly associated with lipid-laden macrophages. Unlike CHI1, GPNMB, is also elevated in mouse models of GD and NPC and can thus be used as a cross-species foam cell marker that could be instrumental in monitoring disease burden in LSD [33,103]. 

In summary, like CHIT1, GPNMB is strongly associated with lipid-laden macrophages. GPNMB, unlike CHIT1, is also elevated in mouse models of GD and NPC. 

## 5. GPNMB: Properties

Human GPNMB is a type 1 transmembrane glycoprotein that, as the result of alternative splicing, occurs as two polypeptide isoforms, one of 572 amino acids and a shorter of 560 amino acids [106,107]. The protein is encoded by the *GPNMB* gene at locus 7p15. Murine GPNMB shares 71% sequence homology with the human orthologue and is slightly smaller (574 amino acids) [108,109]. GPNMB is highly glycosylated: there are twelve putative glycosylation sites in the predicted extracellular part of human protein and eleven in that of the murine orthologue. Several domains in the GPNMB protein have been identified, including an integrin-recognition (RGD) motif and a polycystic kidney disease (PKD)/Chitinase domain in the extracellular part and an immunoreceptor tyrosine-based activation-like motif (ITAM-like; YxxI) and a lysosomal targeting (dileucine) motif in the intracellular part (Figure 1). Extensive N-glycosylation of GPNMB increases its molecular mass to about 120 kDa [110]. After traversing the Golgi apparatus, GPNMB is directed to the cell membrane. At the cell surface, a soluble fragment may be proteolytically released by ADAM-10. Alternatively, GPNMB may be internalized to intracellular vesicles through phagocytosis/endocytosis [111,112,113,114].

GPNMB was originally discovered in a melanoma cell line [115] and occurs in various tissues and cell types. It has relatively high expression in retina and skin, followed by adipose tissue, bone marrow, lung, cervix and immune system, and to lesser extent liver and muscle [116]. Several cell types are reported to express GPNMB: these include phagocytes (dendritic cells and macrophages), osteoclasts and melanocytes [109,117,118,119]. In addition, expression of GPNMB is well documented in melanoma cells as well as other types of cancer cells (reviewed in [120]). 

As addressed in more detail below, GPNMB has been associated with endosomal/lysosomal structures in phagocytes overexpressing the protein during specific stress conditions [113,117,119]. In melanocytes, GPNMB is also targeted to a lysosome-like organelle, the membrane of melanosomes. This particular targeting in melanocytes relies on C-terminal motives in the cytoplasmic tail, shared with the homologous protein premelanosome protein 17 (PMEL17) [114,121]. GPNMB is important in melanosome formation as is reflected by defective formation of pigment by iris pigment epithelium in a mouse strain (DBA/2J (D2)) with a truncated version of Gpnmb [122,123,124]. In humans, a truncated version of GPNMB is associated with hyper- and hypopigmentation of the skin in an autosomal recessive variant of Amyloidosis cutis dyschromica (ACD) [125]. Unlike its homologue PMEL17, GPNMB expression is not restricted to melanocytes. GPNMB has received multiple names. Within the context of bone marrow cells, human GPNMB was initially called hematopoietic growth factor inducible neurokinin-1 type (HGFIN) [126]. In mice, GPNMB was independently identified in dentritic (Langerhans) cells and was named DC-associated, HSPG-dependent integrin ligand (DC-HIL) [109]. This variant shared 88.3% homology to its rat homologue, named osteoactivin [127]. GPNMB was found to be upregulated upon differentiation of monocytes into dendritic cells (DCs), macrophages, and osteoclasts [109,117,118,119]. 

An established regulator of *GPNMB* expression is melanogenesis associated transcription factor (MITF) [119,128,129,130,131,132]. Of note, MITF is a member of the MiT/TFE subfamily of transcription factors known to regulate expression of proteins involved in autophagy and lysosome biogenesis [133,134,135]. Other members of the Mi/TF subfamily are transcription factor EB (TFEB), transcription factor E3 (TFE3), and transcription factor EC (TFEC). Homozygosity for many mutations in *Mitf* alleles gives rise to dysfunctional melanocyte differentiation and defective development of retinal pigment epithelium [136]. Following activation, MITF translocates into the nucleus and binds preferentially to the conserved M-box sequence TCATGTG [129,137,138]. Recent advances in the field of lysosomes have placed the Mi/TFE subfamily at the center of lysosomal homeostasis [133,135,139,140]. The transcriptional activity of TFEB, MITF, and TFE3 can be induced upon pharmacological disruption of lysosomal integrity in cultured cells.

## 6. Function of GPNMB in Myeloid Cells

Many studies on the function of GPNMB in myeloid cells have been performed with DCs. Upon stimulation with interleukin 10 (IL-10), GPNMB expression is found to be induced in DCs through inhibition of phosphoinositide 3-kinase (PI3K)/ RAC-alpha serine/threonine-protein kinase (AKT) and subsequent activation of glycogen synthase kinase-3-β (GSK3β). GSK3 in turn activates MITF to promote expression of GPNMB [131,141]. DC-expressed, membrane bound GPNMB is found to bind to T-cells, thereby inhibiting the proliferation of CD4^+^ and CD8^+^ T-cells and the secretion of IL-2 [142]. Syndecan-4, an heparan sulfate proteoglycan (HSPG) containing membrane protein on activated T-cells, has been identified as primary ligand for GPNMB [143,144,145]. Binding of GPNMB to syndecan-4 is thought to take place in two steps: initial binding via the extracellular arginylglycylaspartic acid (RGD-) domain facilitates PKD-dependent binding [109]. Since the RGD-domain is known to interact with integrin, GPNMB possibly exerts its adhesive action through the activation of integrin interactions [109,146,147,148]. Similarly, DC expressed GPNMB has been reported to bind to dermatophytic fungi in a heparan sulfate dependent manner [Chung 2009]. Another identified binding partner of GPNMB is CD44. Macrophages with anti-inflammatory characteristics (M2) show a marked upregulation of GPNMB [149]. Upon skin wounding, GPNMB derived from infiltrating macrophages was found to promote recruitment of MCSs and subsequent wound repair [150]. Given the fact that MSCs can differentiate into osteoblasts, these studies are in line with findings correlating GPNMB with osteogenesis and osteoblast maturation [33,151,152]. Lastly, GPNMB was found to bind to calnexin, which was suggested to reduce oxidative stress [153].

In several studies on tissue damage, an increase in GPNMB has been reported [154,155,156,157,158,159,160,161,162,163]. Upon renal and liver tissue damage, upregulation of GPNMB is associated with the infiltration of macrophages into the damaged tissue [160,161,162,163]. Interestingly, in a model of reversible liver fibrosis, a subset of profibrotic macrophages (Ly6C^hi^) undergoes a phenotypic switch into macrophages associated with resolution of fibrosis (Ly6C^low^) and concomitantly with increased expression of GPNMB [158]. The phenotypic switch gives rise to macrophages with pro-inflammatory (M1) as well as M2 characteristics and can be triggered by phagocytosis. Of note, a study revealed that GPNMB is crucial for clearance of cellular debris by F4/80^+^ macrophages upon repair of ischemia reperfusion injury (IRI) in the murine kidney [113]. Li et al. showed that GPNMB is associated with LC-3 positive phagocytic vesicles formed upon engulfment of apoptotic cells by macrophages [113]. Monocyte expressed GPNMB seems associated with formation of intracellular vesicles such as (auto-) phagosomes and lysosomes [113,117,119]. 

An M2-phenotype nature of GPNMB positive macrophages is in line with earlier work on splenic Gaucher cells [71]. Morphologically, the Gaucher cell exhibits a foamy appearance due to dramatic enlargement of the lysosomal compartment, in which lipids accumulate in tubular deposits [164]. Gaucher cells are M2-like cells [71] and are surrounded in tissue lesions by macrophages expressing pro-inflammatory molecules such as IL-1β or monocyte chemoattractant protein 1 (MCP-1) [71]. Possibly, the latter cells are responsible for the elevated levels of the chemokines MIP-1α and MIP-1β in plasma of symptomatic Gaucher patients [165]. 

## 7. GPNMB and Foam Cells in Acquired ‘Metabolic’ Disorders

As indicated earlier, defects in the lysosomal catabolic machinery trigger massive induction of GPNMB in macrophages in spleen, liver, and brain in GD and NPC [33,97,101,103,104]. Interestingly, when the amount of lipid substrate exceeds the lysosomal capacity in macrophages, a foamy appearance and clear induction of GPNMB is observed [132,153,166,167]. Examples of this include cholesterol accumulation in atherosclerosis, lipid accumulation in macrophages during obesity, and myelin accumulation in brain macrophages during multiple sclerosis (MS). In a proteome analysis of ascending aortic extracts of rabbits fed a high cholesterol diet (HCD), 15-fold elevated GPNMB levels were detected [168]. In LDLR, ^-/-^ mice fed a HCD a 300-fold induction of *Gpnmb* was found in liver, most likely in Kupffer cells [166]. Interestingly, GPNMB was also found to be increased in human subjects with fatty liver disease. In subjects with non-alcoholic steatohepatitis, plasma GPNMB levels were significantly elevated compared to simple steatosis [153]. Studies on rodent models of obesity, leptin-deficient, and high fat diet fed mice, revealed striking induction of GPNMB in obese adipose tissue macrophages [132]. Again, a high lipid load derived from phagocytosis of dysfunctional/apoptotic adipocytes is the likely trigger. In the liver, a less pronounced induction of GPNMB was detected in Kupffer cells. Consistently, increased lysosomal volume occurs in obese adipose macrophages [169]. Also in human obese adipose tissue, *GPNMB* expression was found to be increased [132]. In post-mortem analyzed human brain tissue of MS patients, it was found that *GPNMB* is increased around the rim of chronic active lesions. This rim is characterized by the abundant presence of foamy, lipid-laden, macrophages [167]. The *GPNMB* increase was accompanied by an increase in macrophage restricted *CD68* expression, as well as *CHIT* and *CCL18*. Together these data point to a role of accumulating lipids like (glyco) sphingolipids and cholesterol as inducers of GPNMB. During an LSD, flaws in the catabolic machinery in macrophages drive lipid accumulation, whereas in acquired metabolic diseases such as atherosclerosis and obesity, as well as MS, the lysosomal load of lipids exceeds the catabolic capacity.

In vitro studies support a connection between GPNMB and lysosomal function. A variety of lysosomal stressors, including sucrose, chloroquine, bafilomycin, concanamycin A, and palmitate (but not oleate), induce GPNMB expression in cultured RAW264.7 cells [132,170]. Upregulation of GPNMB occurs also in RAW264.7 macrophages upon blocking cholesterol efflux from the lysosome by U18666A, thereby mimicking aspects of NPC pathology [103]. Impairing lysosomal function in different ways (increasing lumenal pH, swelling by accumulation of non-degradable material, excessive lipid load and impaired lipid efflux) all induces upregulation of GPNMB. mTORC1 is known to mediate regulation of lysosome biogenesis and autophagy via the Mi/TFE transcription factors [132,171]. Consistently, inhibition of mTORC1 activity with torin 1 induces markedly GPNMB [149]. Recently, the buffer HEPES was found to also potently induce GPNMB expression through Mi/TFE members in cultured RAW264.7 cells [170]. In this manner the presence of HEPES impacts on cellular lysosomal enzyme levels. Therefore, the finding highlights the importance of culture conditions (such as presence of HEPES) for diagnosis of LSDs with cultured cells. 

Besides being highly expressed in macrophages in LSDs and acquired metabolic disorders, GPNMB is also increasingly linked to neuroinflammation [172,173,174]. For example, elevated GPNMB in glioma tissue stems largely from reactive glioma-associated phagocytosing microglia and macrophages (GAMs) [175,176,177,178]. Data also link GPNMB to neurodegeneration, including cerebral ischemia, amyotrophic lateral sclerosis (ALS), Alzheimers Disease (AD), Multiple Sclerosis (MS), and Parkinson Disease (PD) [179,180,181,182,183,184]. Increased *GPNMB* expression has been associated with a particular microglial state called the ‘microglial neurodegenerative phenotype’(MGnD), observed in mouse models for AD, MS, and ALS [185]. This phenotype was shown to markedly differ from M1-differentiated microglia and cells with this phenotype were associated with amyloid-β deposits in a murine AD-model [182,185]. Strikingly, upon injection of apoptotic neurons in the hippocampus and cortex of healthy mice, the MGnD-phenotype could be induced through TREM2, a phosphatidylserine sensing protein, and upregulation of apolipoprotein E (APOE). Elevated levels of GPNMB were also found in the substantia nigra (SN) of PD-patients [183,184]. Moloney et al. could recapitulate this GPNMB-increase in mice by blocking GBA activity through systemic conduritol-beta-epoxide administration, which suggests a connection between neuronopathic glycosphingolipidoses and PD [38,97,103,184,186]. In a chemically induced mouse model of PD, CD44 has been proposed to function as binding partner of GPNMB in the SN [183]. The dopamine-producing neurons in the SN produce neuromelanin, causing their pigmentation. Neuromelanin increases upon ageing and has been associated with PD. Neuromelanin accumulation may occur along with defective trafficking and degradation by the endolysosomal apparatus [187,188]. It is conceivable that GPNMB is upregulated as response to lysosomal stress caused by accumulating, undegradable neuromelanin.

It is of interest to consider the advantages and disadvantages of the use of GPNMB as marker of lipid-laden macrophages, instead of chitotriosidase or CCL18. Firstly, GPNMB can be conveniently quantified by ELISA, a methodology accessible to most laboratories. Secondly, GPNMB is expressed also by lipid-laden macrophages in mice; this is not the case for either chitotriosidase or CCL18 [73,85]. A potential disadvantage is the present lack of knowledge of possible genetic heterogeneity in (expression of) *GPNMB*. This may not be irrelevant: for example, the *CHIT1* gene has common mutations, resulting in no protein or enzyme with abnormal catalytic features [67,73]. This limits the value of CHIT1 as marker of lipid-laden macrophages. The selectivity of GPNMB as marker warrants further research. It is still unclear to which extent other cell types than lipid-laden macrophages may also express and secrete GPNMB during pathological conditions. It seems likely that in disease characterized by the presence of lipid-laden macrophages abnormalities in GPNMB will occur: such candidate diseases include Wolman disease and the more benign mature variant, cholesteryl ester storage disorder, both caused by a deficiency in lysosomal acid lipase [189]. In this disorder chitotriosidase is also markedly elevated [190].

## 8. Conclusions

Lipid-laden macrophages may orchestrate pathology, an accepted notion in the field of inborn lysosomal storage disorders and more recently also in the field of the metabolic syndrome (Figure 2). The development of ERT for specific LSDs has led in the last decades to the identification of markers of lipid-laden macrophages. In LSDs characterized by foamy macrophages as storage cells, plasma GPNMB has been shown to accurately reflect disease burden. Moreover, GPNMB is also applicable in mouse models of LSDs like GD and NPC. GPNMB is also increased in several acquired diseases, such as the metabolic syndrome and neurodegeneration. It therefore might be that specific LSDs and the latter disease conditions share elements in pathophysiology, in particular the involvement of accumulating foamy, lysosomal stressed, macrophages (see Figure 2). 

GPNMB is among the highest upregulated proteins in lipid-laden macrophages. Nevertheless, at present its exact function in the foamy macrophage remains largely enigmatic. Important unanswered questions concern the function(s) served by GPNMB, either the cellular membrane-bound or (extracellular) soluble isoforms, in lipid-laden macrophages and beyond. 

## Figures and Tables

**Figure 1 ijms-20-00066-f001:**
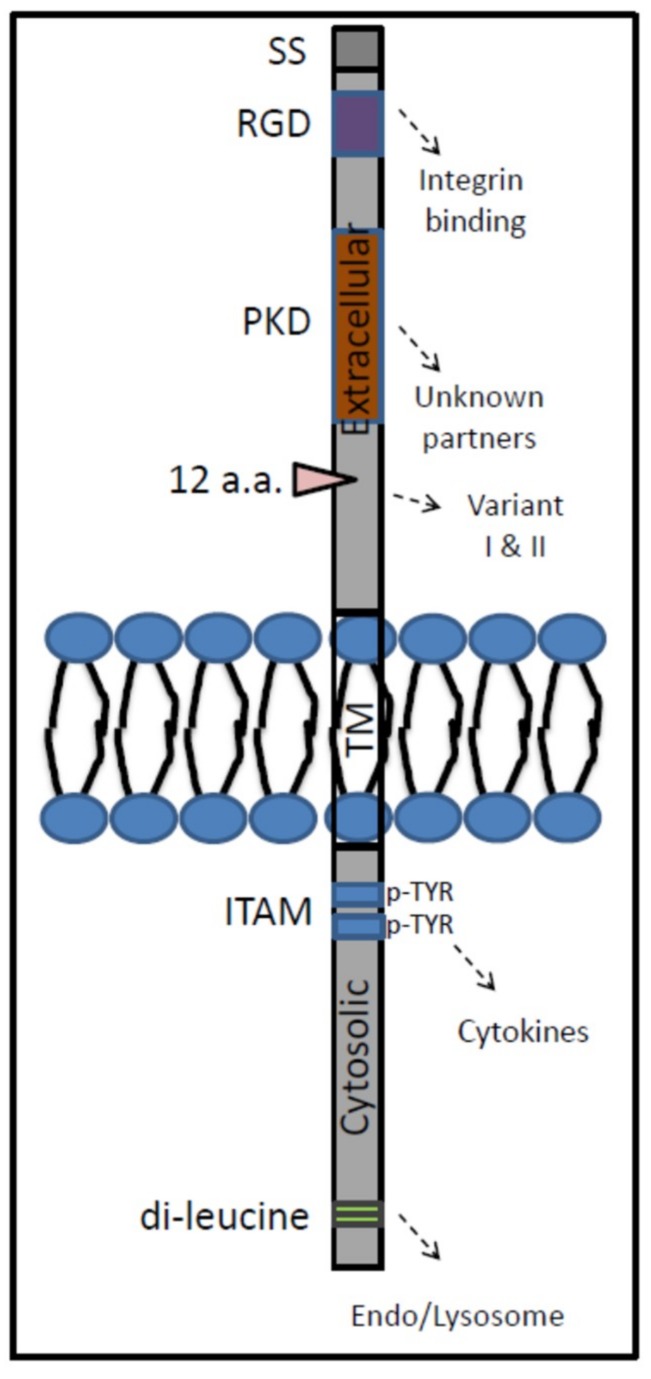
Schematic overview of Gpnmb protein. SS, signal sequence; RGD, RGD tripeptide; PKD, Polycystic kidney disease domain; a.a., amino acid; ADAM; a disintegrin and metalloproteinase; ITAM, immunoreceptor tyrosine-based activation-like motif; TM, transmembrane domain.

**Figure 2 ijms-20-00066-f002:**
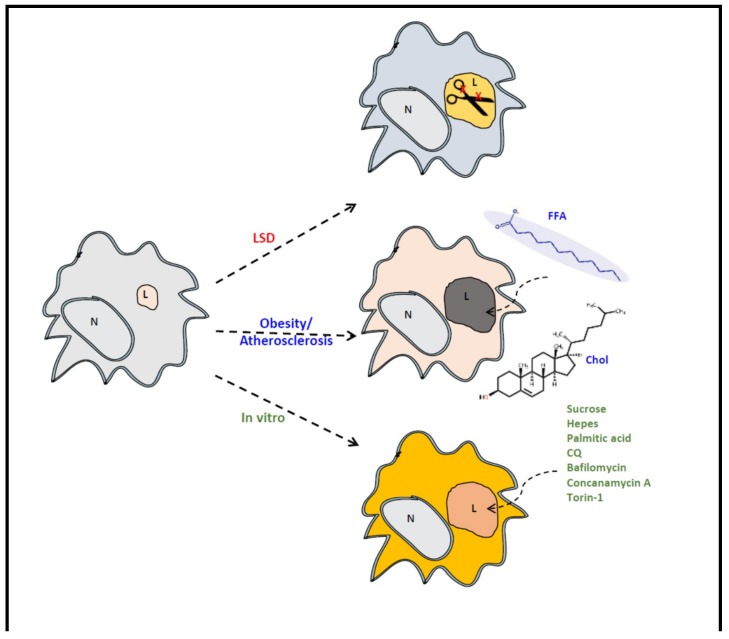
Model for lysosomal dysfunction in LSD, metabolic syndrome and cultured cells; lysosomal dysfunction could be caused in vivo by deficiencies in lysosomal hydrolases (LSD) or chronic excess of nutritional intake (metabolic syndrome). In vitro, lysosomal dysfunction can be recapitulated by several compounds that model in vivo systems. FFA, free fatty acid; Chol, cholesterol; CQ, chloroquine; N, nucleus; L, lysosome.

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
