# Peer review of "Glycoprotein Non-Metastatic Protein B: An Emerging Biomarker for Lysosomal Dysfunction in Macrophages"

_ijms, 2018, doi:10.3390/ijms20010066_

Reviewer 1 Report

This paper gives an interesting new view about biomarkers in lysosomal storage disorders and in oder disorders related to lysosomal function. I believe it is well written and deserve publication.

Author Response

Dear reviewer, 

Many thanks for the kind judgement. 

With best regards,

 Also on behalf of the other co-authors,

Martijn van der Lienden

Reviewer 2 Report

The review about the biomarkers The review related to biomarkers in lysosomal disorders entitled "Glycoprotein non metastatic protein B: An Emerging Biomarker for Lysosomal Dysfunction in Macrophages" constitute a very complete work. Nevertheless are missing other markers of inflammation present in some of the lysosomal diseases such as ferritin in Gaucher disease or the angiotensin converting enzyme in the section on proteins.  

The reference to oxidation products such as plasma oxysterols that are increased in disorders of cholesterol could be expanded somewhat more since it is a good marker of accumulation of t could be expanded somewhat more since it is a good marker of accumulation ofaccumulation. It would be good if in theconclusions the qualities of the proposed GPNMB marker and advantages / disadvantages in relation to existing markers are explained in more detail. since it is a good marker of accumulation of intracellular cholesterol. Taking into account that the role of GPNMB is referred to in obesity and tissue lipid  accumulations, and then could be a useful marker in the lysosomal acid lipase deficiency.

It would be good if in the conclusions  section to add more arguments about the qualities of the proposed GPNMB marker and advantages / disadvantagesin relation to existing markers in order to provide an opinion.

Author Response

Dear reviewer,

Many thanks for the swift review of our manuscript. We have adjusted our manuscript according to the following suggestions and hope we have sufficiently answered your questions.

Point 1: There are missing other markers of inflammation present in some of the lysosomal diseases such as ferritin in Gaucher disease or the angiotensin converting enzyme in the section on proteins.

Response 1: We have added information on biochemical abnormalities in Gaucher patients including ferritin and ACE (page 3). The addition is as follows: 

 'Already reported were a number of plasma protein abnormalities in Gaucher patients, for example elevated levels of lysozyme, beta-hexosaminidase, ferritin, tartrate-resistant acid phosphatase (TRAP) and angiotensin-converting enzyme (ACE), see for a review [64]. However, for none of these abnormalities it was clear that they are uniquely related to Gaucher cells and not also released by other cell types, as for example TRAP by pro-inflammatory macrophages, osteoclasts and dendritic cells [65].'

Point 2: The reference to oxidation products such as plasma oxysterols that are increased in disorders of cholesterol could be expanded somewhat more since it is a good marker of accumulation.

Response 2: We spend more attention to oxidized cholesterol metabolite abnormalities in relation to NPC, including further references (page 3), as follows:

'Oxysterols are formed in the body through enzymatic, and non-enzymatic reactions involving reactive oxygen species (ROS). The latter reaction seems to be driving the enhanced levels of oxysterols in NPC [46–50]. Moderate elevation of oxysterol levels is also observed in other cholesterol related storage diseases such as atherosclerosis, obesity and diabetes [51–53].' 

Point 3: It would be good if in the conclusions the qualities of the proposed GPNMB marker and advantages / disadvantages in relation to existing markers are explained in more detail.since it is a good marker of accumulation of intracellular cholesterol. Taking into account that the role of GPNMB is referred to in obesity and tissue lipid accumulations, and then could be a useful marker in the lysosomal acid lipase deficiency. It would be good if in the conclusions  section to add more arguments about the qualities of the proposed GPNMB marker and advantages / disadvantages in relation to existing markers in order to provide an opinion.

Response 3: We elaborate more on the advantages/disadvantages of GPNMB as marker of lipid-laden macrophages (page 8). In connection to this we also hypothesize that some other disease conditions, like acid lipase deficiency (Wolmann disease) might be associated with GPNMB abnormalities: 

'It is of interest to consider the advantages and disadvantages of the use of GPNMB as marker of lipid laden macrophages, instead of chitotriosidase or CCL18. Firstly, GPNMB can be conveniently quantified by ELISA, a methodology accessible to most laboratories. Secondly, GPNMB is expressed also by lipid laden macrophages in mice; this is not the case for either chitriosidase or CCL18 [74,86]. A potential disadvantage is the present lack of knowledge on possible genetic heterogeneity in (expression of) GPNMB. This may not be irrelevant: for example, the CHIT1 gene has common mutations, resulting in no protein or enzyme with abnormal catalytic features [68,74]. This limits the value of CHIT1 as marker of lipid laden macrophages. The selectivity of GPNMB as marker warrants further research. It is still unclear to which extent other cell types than lipid-laden macrophages may also express and secrete GPNMB during pathological conditions. It seems likely that in disease characterized by the presence of lipid laden macrophages abnormalities in GPNMB will occur: such candidate diseases include Wolman disease and the more benign mature variant, cholesteryl ester storage disorder, both caused by a deficiency in lysosomal acid lipase [190]. In this disorder chitotriosidase is also markedly elevated[191].'

 With best regards,

Also on behalf of the co-authors,

Martijn van der Lienden